# A Broadband Power Amplifier Based on a Novel Filter Matching Network

**Zheng Li \*, Jingchang Nan, Mingming Gao and Yun Niu**

School of Electronics and Information Engineering, Liaoning Technical University, Huludao 125100, China
* Correspondence: lizheng20sui@163.com

**Abstract:** This paper presents a new realization method of a broadband power amplifier based on a novel filter matching network. The novel matching network based on band-pass filter has an excellent frequency-selection function, which can ensure the novel matching network has excellent characteristics in the aim band and generates the out-of-band harmonic suppression. Finally, we manufactured the power amplifier and measured it. The saturated output power is greater than 40 dBm in the range of 1 to 3 GHz, limited to ±1.5 dB of the gain flatness, and the rejection of harmonic is stronger than −20 dBc.

**Keywords:** broadband; gain flatness; negative feedback; filter network; Norton transform

## 1. Introduction

The power amplifier (PA) has the function of converting DC energy into ~~RF~~ radio frequency (RF) signal energy, which plays an extremely significant role in wireless communication system. With the popularization of the 5G technology and the advancement of the 6G technology, the operating mode and communication standard of the communication system will become more and more diversified [1]. Broadband communication is the key to address the above problems [2]. Thus, the research on broadband PAs has a strong practical value.

We compared several novel broadband PA design approaches. The reference [3] presents a 28 GHz wideband PA with the dual-pole tuning superposition technique. The PA employs a four-stage pseudo-differential structure together with the two-way combining approach to enhance the power gain. We have to admit that this method although the performance is very good, but can only be applied to several special cases, its practicability is weak. In reference [4], a broadband Class-E PA was realized using a Varactor Diode Stack. Bandwidth and linearity have always been criticized for switching PAs. So only through other means to try to broaden the bandwidth, but this will make the design process complicated, the size of the circuit is too large. Therefore, switching PA and continuous switching PA are not the first choice for designing wideband PA in engineering practice; In reference [5], based on the step impedance converter and load-pull, the matching networks were designed, so as to realize the design of broadband PA, but the bandwidth that ordinary step impedance converter can achieve is limited after all; Additionally, reference [6] proposed a type of elliptic low-pass and band-pass matching network. A PA was realized from 1.45 to 2.55 GHz. Nevertheless, the inherent poles of the matching networks limit the bandwidth of the PA.

Although some valuable advances have been obtained in existing research [7–11], the design of broadband PAs is often constrained by drastic changes in impedance values and misfits between fixed matching networks [12]. In this paper, we design a broadband PA, using filter matching networks with large impedance variation space. It is able to convert the rapidly changing impedance in a large range to around 50 Ω. This ability is

superior to other existing studies. The designed PA has wider bandwidth and excellent harmonic control capability, which can work at 1 GHz to 3 GHz, with good gain flatness and output capacity above 40 dBm. In general, the proposed design method in this paper can break through the limitations of volume, complexity and applicability. It can broaden the bandwidth of PA greatly and show excellent harmonic suppression ability. This makes the designed PA perform well both in-band and out-of-band.

## 2. Proposed Network Topology

In this article, we use a GaN transistor to design the broadband PA. One of the most widely accepted GaN transistor model is shown in Figure 1. A higher order LC-loop sections is usually needed by a wider band, for a broadband matching network design, which means the low-pass form of parasitic parameters can be easily absorbed into the design of matching network and extend the band. Therefore, a matching network based on low-pass filter is extremely appropriate for the output term of the transistor based on the equivalent circuit of GaN transistors.

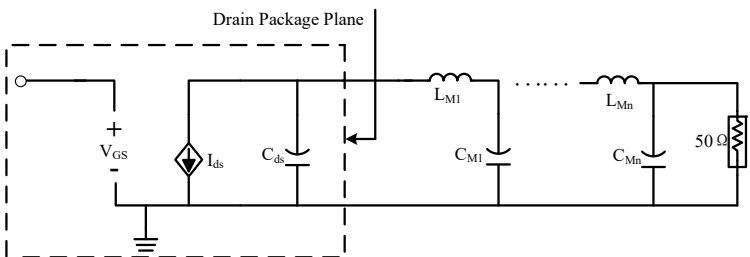

**Figure 1.** Transistor equivalent model and output equivalent circuit. [13].

Thus, the impedance of the transistor is complex. And the matching network converts the impedance to a specific value [14]. And the imaginary part of the impedance is usually considered to be provided by the capacitor [15,16]. In the low-pass filter prototype shown in Figure 2, $R_1$ represents the real resistance part of the port impedance, while the imaginary part is provided by the shunt capacitor $C_1$, which form an equivalent to the impedance value of the transistor port [17,18]. The equivalent impedance is converted into real resistance $R_2$ under the action of $L_1$ and $C_2$.

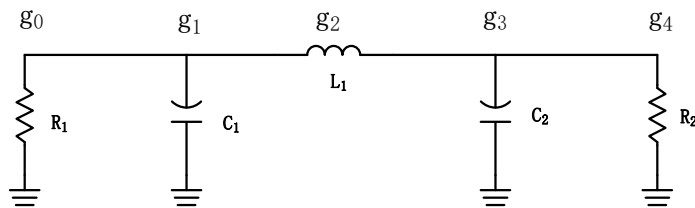

**Figure 2.** Structure of the low-pass filter prototype [19].

Dale E. Dawson, based on the Bode-Fano criterion, obtained the solution method for the value of g and the remaining components' value under the condition of fixed $R_1$ and $C_1$:

$$g_0 = 1 \tag{1}$$

$$g_1 = Q\omega / g_0 \tag{2}$$

$$g_j = 1 / g_{j-1}(K_{j-1,j})^2, j \geq 2 \tag{3}$$

$$g_{n+1} = Q\omega / D \cdot g_n \tag{4}$$

$$g_1 = Q\omega / g_0 \tag{2}$$

In the above equation, where Q denotes the quality factor of circuit; n denotes the order of filter network; K denotes fixed insertion loss coefficient; D denotes a discriminator to judge whether the above equations have roots. Moreover, $\omega_0 = \sqrt{\omega_1 \omega_2}$ ; $\omega = (\omega_2 - \omega_1) / \omega_0$. Here, $\omega$ represents the angular frequency corresponding to the upper and lower edge frequency points of the operating band, respectively.

$$Q = R_1 \omega_0 C_1 \tag{6}$$

$$\theta = \pi / 2n \tag{7}$$

$$D = \frac{\sinh a}{(\frac{1}{Q\omega})\sin(\frac{\pi}{2n})} - 1 \tag{8}$$

$$\sinh a = \sqrt{r_n + (c/2)^2} + c/2 \tag{9}$$

Especially, when solving $\sinh a$, the value of $r_n$ in Equations (5) and (9) varies according to the matching network parameters $Q$, $\omega$ and $n$. The $r_n$ is the relationship coefficient between the minimum ripple and insertion loss of the matching network. The table in Reference [20] lists the values of $r_n$ under different conditions. It can be seen from the table that $r_n$ is equal to 0.33.

In order to have a better out-of-band rejection at both high and low frequencies, the low-pass filter prototype is transformed into a Band-Pass filter model [21]. Under the premise that $R_1$ and $C_1$ are fixed, the remaining components are converted into $LC$ resonant circuit. According to Equations (10) and (11):

$$\omega_c L = \omega_2 L - \frac{1}{\omega_2 C} = \omega_0 L(\frac{\omega_2}{\omega_0} - \frac{\omega_0}{\omega_2}) \tag{10}$$

$$\omega_c C = \omega_1 L - \frac{1}{\omega_1 C} = \omega_0 L(\frac{\omega_1}{\omega_0} - \frac{\omega_0}{\omega_1}) \tag{11}$$

where $\omega$ denotes the low-pass corner frequency, and $\omega_C = \omega_1 - \omega_2$, $\omega_0 = 1/\sqrt{LC} = \sqrt{\omega_1 \omega_2}$, separately. The structure of the converted Band-Pass filter model is shown in Figure 3.

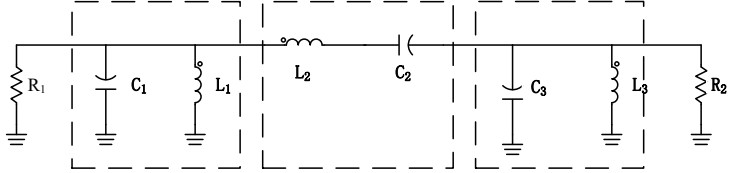

**Figure 3.** The structure of Band-Pass filter model.

$R_2$ is required to be designed as 50 Ω for application in communication system. However, $R_2$ cannot be guaranteed to be constant at 50 Ω, so a Norton transformation is required. The Norton transformation is shown in Figure 4 [22-24]. Based on the method given in Figure 4, the circuit structure is adjusted at $C_2$ and $C_3$ in Figure 3. $R_1$ and $C_1$ are

still used to be equivalent to the complex port impedance of the transistor. The subsequent circuit structure can convert the changed complex impedance into real-impedance. Through the adjustment of the circuit structure, the impedance that changes sharply with the frequency can be converted to around 50 Ω in a wide bandwidth, which means that the circuit can provide a large impedance conversion space to achieve the performance of broadband. The topology of the matching network is shown in Figure 5.

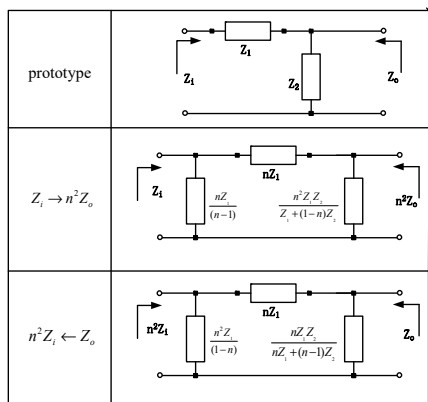

**Figure 4.** The process of Norton transformation.

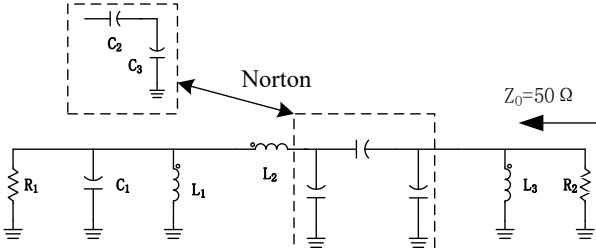

**Figure 5.** The topology of matching network.

In summary, except for the special case that $C_1$ is very large and $R_1$ is very small, the matching network can convert the impedance of most transistors at any frequency to 50 Ω, and it can achieve the function only through the third-order loops. Since the proposed topology is based on the band-pass filter theory, the aim band and bandwidth can be designed manually. Beyond the aim frequency band, the matching network can attenuate signals greatly. On the one hand, it can increase the isolation between the different frequency signals to avoid cross-talk; on the other hand, it can suppress the power consumption of harmonic components and improve efficiency. Theoretically, the bandwidth that can be achieved by this topology which can be controlled manually, and there are no obvious restrictions. In addition, the matching network can adapt to the situation that the impedance value changes with the frequency sharply. In all, the matching network provides a larger impedance variation space than other researches.

### 3. Design of Proposed Broadband PA

*3.1. The Design of PA*

The design was manufactured on a FR4 substrate, and the dielectric constant is 4.5. Meanwhile, the plate thickness is 0.8mm, and the transistor used in the PA is Cree's CGH40010F GaN transistors. The drain bias voltage is 28 V, and the gate bias voltage is −2.5 V, leading to the PA was biased in class AB. Since the gain of transistors has the characteristic of frequency roll-off, the difference of gain between high and low frequencies is large. Therefore, as shown in Figure 6, a negative feedback network is built between the gate and the drain, so that the PA obtains a flat gain over the entire band. The negative feedback network consists of resistors, capacitors and inductors.

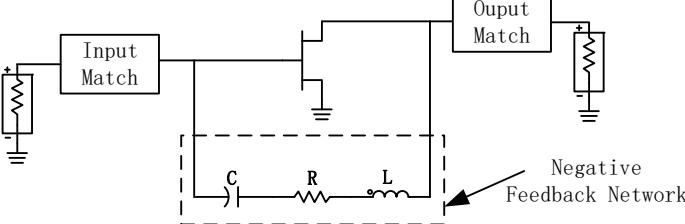

**Figure 6.** The structure of negative feedback network.

The value of the resistance is affected by the gain and bandwidth. The role of the inductance is to reduce the feedback of high-frequency signals, so as to obtain a better gain flatness. The capacitor plays a role of isolating DC, and it can adjust the phase of the feedback signal. The gain of the PA in the whole operating band can be maintained at a stable level by using the inductance to adjust the feedback signals of different frequencies. The above three circuit components cooperate with each other and adjust their values to obtain the best gain flatness. It is worth mentioning that the negative feedback circuit is good for improving the input and output standing-wave ratio and the stability of the circuit. Compared with balance structure, reactance matching, distributed structure and other methods which has similar function, negative feedback circuit has the advantages of small size, simple structure and insensitive circuit components. Therefore, the negative feedback circuit becomes the preferred method for PCB PA to obtain good gain flatness.

Before designing the matching network, the impedance value needs to be selected by load-pull. The impedance values of transistor CGH40010F gate and drain at different frequency points are shown in Table 1.

**Table 1.** The optimal impedance at different frequency points.

| Freq/GHz | $Z_{load}/\Omega$ | Freq/GHz | $Z_{source}/\Omega$ |
|---|---|---|---|
| 1.0 | 27.54 + j × 9.87 | 1.0 | 20.74+j × 15.36 |
| 1.5 | 25.8 + j × 16.1 | 1.5 | 20.74+j × 15.36 |
| 2.0 | 21.34 + j × 8.35 | 2.0 | 8.77 + j × 10.45 |
| 2.5 | 15.53 + j × 11.94 | 2.5 | 6.66 + j × 3.29 |
| 3.0 | 16.45 + j × 7.15 | 3.0 | 8.03 − j × 1.45 |

The impedance value of the transistor at various frequency points is different. It is necessary to select a group of optimal impedance to design the matching networks, so that, the PA has good matching performance in the whole operating band. The band of this design reaches several octaves, so the effect of harmonics should be considered when selecting impedance. Finally, we choose $Z_{load} = 19.2 + j \times 13.6\Omega$ and $Z_{source} = 8.77 + j \times 10.45\Omega$ as the optimal impedance. Then, the matching networks are shown in Figure 7.

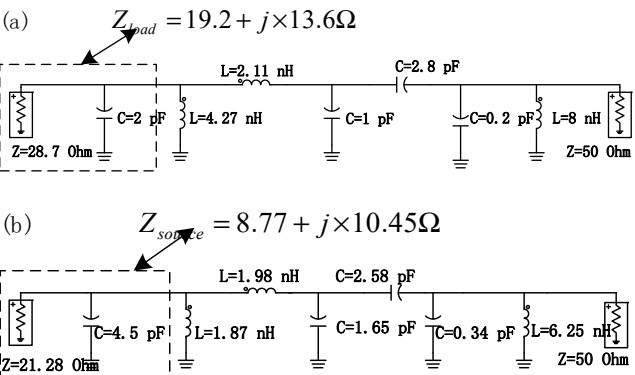

**Figure 7.** The structure of matching networks. (**a**)output matching network; (**b**)input matching network.

As shown in Figure 7, the impedance of the transistor is equated by the resistance and the shunt capacitance.

Figure 8a shows the impedance track of the output matching network in the range of 1~3 GHz, and Figure 8b shows the impedance track of the input matching network in the range of 1~3 GHz. From these Figures, it can be seen that the impedance points of the two matching networks in the range of 1 to 3 GHz are around 50 Ω and maintain a small distance from the 50 Ω point. Compared with the impedance values shown in Table 1, the results verify that the matching network has an excellent ability to transform the varying impedance in a large range. If there is a need for wider bandwidth, it only need to calculate the value of each LC-loop according to the upper and lower frequency limits, so the matching network structure proposed in this paper has stronger scalability and wider applicability than other studies.

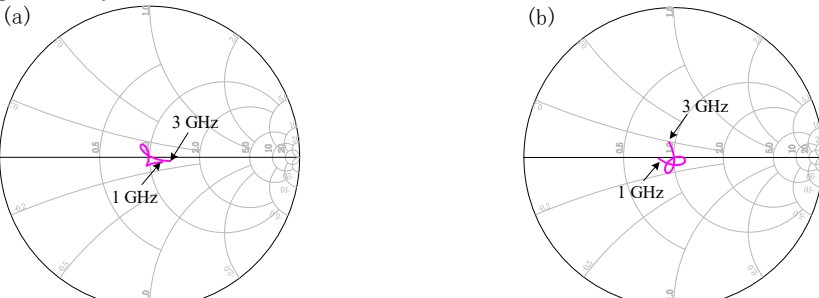

**Figure 8.** The impedance tracks. (**a**) output matching network; (**b**) input matching network.

Then, from the S-parameter simulation results in Figure 9, it can be seen that the two matching networks not only have outstanding matching characteristics in the operating band, but also can produce out-of-band rejection very quickly, which will have a perfect effect on the harmonic suppression.

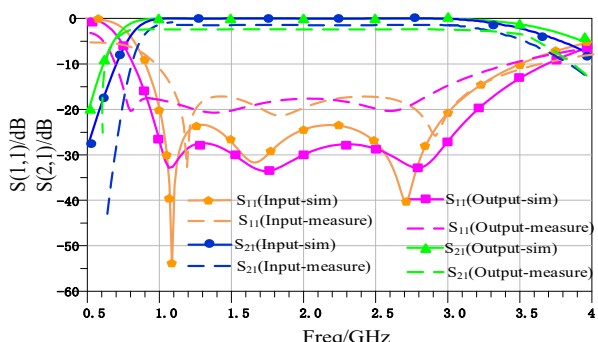

**Figure 9.** S-parameter simulation and measure results.

Thirdly, we transformed the lumped elements into microstrip lines. According to Figure 10, there are two structures to choose from here.

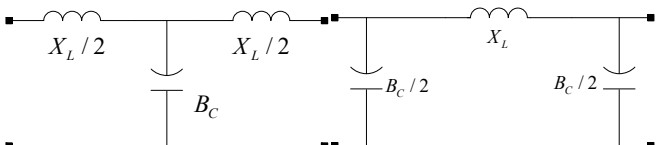

**Figure 10.** T-type and $\pi$-type structure.

Using the transition matrix can prove its equivalence. The transition matrix of the microstrip line can be expressed as:

$$[A]_1 = \begin{bmatrix} \cos\theta & jZ_0\sin\theta \\ \dfrac{j\sin\theta}{Z_0} & \cos\theta \end{bmatrix} \tag{12}$$

$$[A]_T = \begin{bmatrix} 1 - \dfrac{1}{2}X_L B_C & \dfrac{1}{2}jX_L(2 - \dfrac{1}{2}X_L B_C) \\ jB_C & 1 - \dfrac{1}{2}X_L B_C \end{bmatrix} \tag{13}$$

$$[A]_\pi = \begin{bmatrix} 1 - \dfrac{1}{2}X_L B_C & jX_L \\ \dfrac{1}{2}jB_C(2 - \dfrac{1}{2}X_L B_C) & 1 - \dfrac{1}{2}X_L B_C \end{bmatrix} \tag{14}$$

Equations (13) and (14) are established as equations with Equation 12, respectively. Therefore, the equivalent conditions can be expressed as:

$$B_C = Y_0 \sin\frac{2\pi L}{\lambda_g} \tag{16}$$

$$X_L = 2Z_0 tg\frac{\pi L}{\lambda_g} \tag{17}$$

Then, If the inductance is small in the T-shaped structure and the capacitance is small in the $\pi$-shaped structure, it can be appropriately ignored. Individual shunt inductors are equivalent using short-circuited microstrip lines; shunt capacitors are equivalent using open-circuited microstrip lines. Finally, connecting each module, and the overall circuit of the proposed PA is shown in Figure 11.

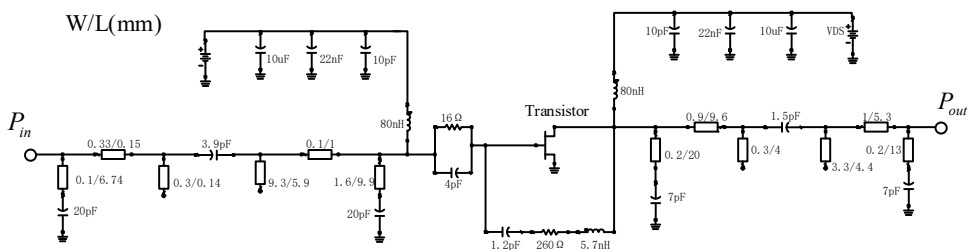

**Figure 11.** The proposed PA circuit.

*3.2. Simulation of Broadband PA*

Since the simulation results are somewhat different from the ideal transmission line after conversion to the actual microstrip line, the microstrip line after the conversion is slightly tuned, so that the designed PA has better performance. In this section, we measured the performance of the proposed PA by performing S-parameter and HB-balance simulations, respectively. Firstly, we performed S-parameter simulations.

According to the simulation results in Figure 12a, $S_{11}$ is less than −10 dB in the whole operating band, indicating that the matching performance is excellent. And the gain flatness is less than 1.2 dB. Figure12b shows the impedance change track of the PA at different frequency points. It can be found that the impedance always keeps a small distance from 50 Ω, and the curve is smooth, which means that the PA does not have negative phenomena, for example, self-excitation. Secondly, we performed HB balance simulations. The results are shown in Figure 13.

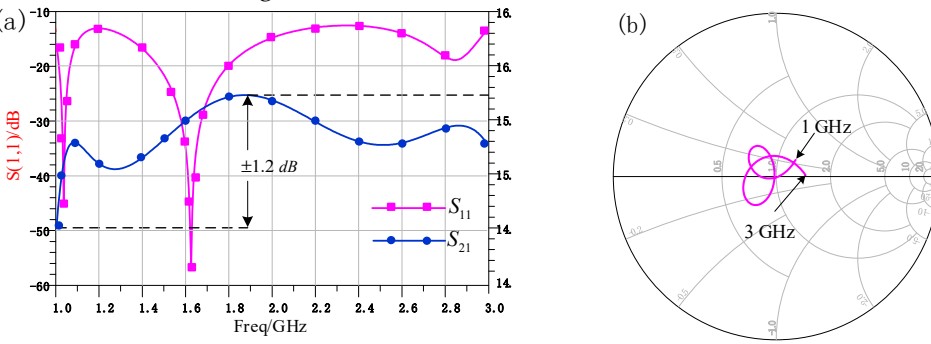

**Figure 12.** S-parameter simulation results of the proposed PA. (**a**)S-parameter simulation results；(**b**)The Impedance track of the proposed PA.

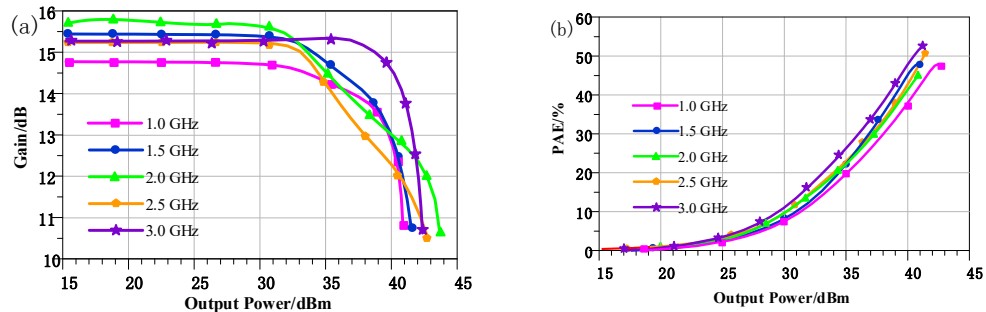

**Figure 13.** HB-balance simulation results of the proposed PA. (**a**)Output power & Gain; (**b**) Output power & PAE.

From the HB-balance simulation results in Figure 13a, it can be seen that the gain is in the range of 14.7~15.8 dB and the gain flatness is less than ±1.1 dB. The saturation output power of the proposed PA is greater than 40dBm. According to Figure 13b, the power added efficiency (PAE) is greater than 45% in the entire operating band. In summary, the proposed PA has excellent broadband characteristics and typical class AB characteristics, with high output capability and good gain flatness.

## 4. Measurement Results

The photograph of the fabricated PA is shown in Figure 14. To verify the simulation results in Section 3, QAM modulated signals used to measure the PA.

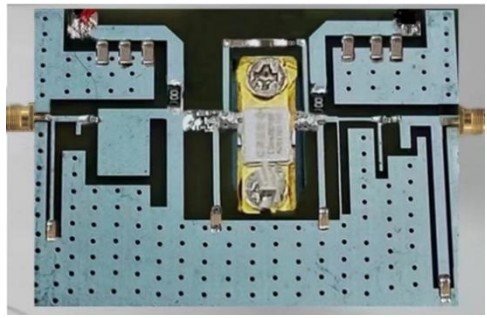

**Figure 14.** The photograph of the fabricated PA.

Firstly, the measured results of the PA at different frequencies for output power & gain and output power & PAE are shown in Figure 15. The Figure 15a shows that the gain remains at a relatively stable level in the operating band, the gain flatness is less than 1.5 dB. And The Figure 15b shows that the PAE is greater than 45% at the saturation output points. The gain simulation results are slightly higher than the measured results, and the PAE results are basically the same as the measured results.

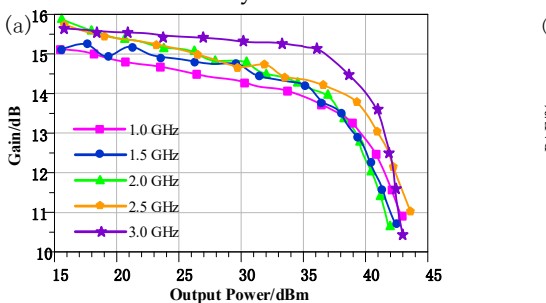 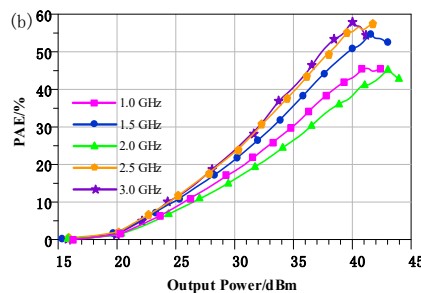

**Figure 15.** The measured results of the proposed PA. (**a**) Output power & Gain; (**b**) Output power & PAE.

Secondly, we also measured the performance of the PA in the frequency domain, and the results are shown in Figure 16, they can also reflect the ability of harmonic suppression. According to Figure 16a, the output power level of the PA is more stable, broadband performance is perfect. Then, a scan of the output spectra was performed. It can be seen from Figure 16b that in the frequency band above 3 GH, the PA attenuates the signal, and the output power spectrum decreases to below −20 dBc. We generally study the second harmonic to fifth harmonic of the PA, which are 2 to 6 GHz, 3 to 9 GHz, 4 to 12 GHz, and 5 to 15 GHz, respectively. The harmonic components of these frequency bands are more than 60 dBc different from the fundamental wave of 1 to 3 GH. At 9.2 GHz, the difference even exceeds 100 dBc. This means that the proposed matching networks have excellent suppression effect on out-of-band harmonics.

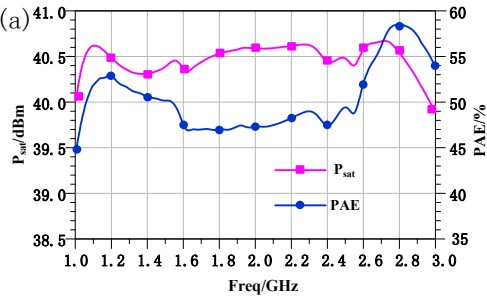 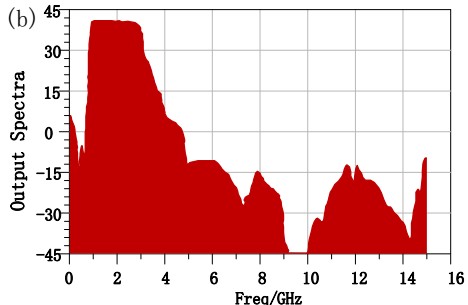

**Figure 16.** The measured results of frequency domain performance. (**a**) $P_{sat}$ & PAE performance; (**b**) Output spectra.

Finally, Table 2 summarizes the performance of some recently reported broadband PAs. To make an efficient comparison, the measurement results of our Broadband PA are also listed. Clearly, the broadband PA structure proposed in this paper has great advantages in terms of bandwidth and harmonic rejection.

**Table 2.** Performances compared with PA proposed in other articles.

| Ref | [25] | [26] | [27] | This Work |
|---|---|---|---|---|
| Transistor | GaN | GaN | GaN | GaN |
| Drain voltage/V | 28 V | 28 V | 28 V | 28 V |
| Class of PA | hybrid modes | continuous | F | AB |
| Bandwidth method | Harmonic tuned | Continuous theory | Reconfigurable | filter matching |
| Band/GHz | 0.8~2.1 | 1.2~2.7 | 0.8~1.8 | 1~3 |
| Output/dBm | 42 | 42.1 | 41.5 | >40 |
| PAE/% | <70 | >40 | <75 | >45 |
| Gain flatness/dB | >4 | >4 | <2.9 | <1.5 |
| Harmonic | >−14 | \ | <−20 | <−20 |

## 5. Conclusions

In this paper, a new design method for broadband PAs is proposed. Firstly, a novel filter matching network is proposed based on the theory of the Band-Pass filter. By relying on the matching network, the PA has excellent broadband performance and strong harmonic rejection, which plays an important role in improving the efficiency and stability of the PA. And the matching network provides a larger impedance variation space than other researches, this is the reason why it can convert the impedance of a large range of changes to around 50 Ω. By recalculating the value of each device, it can be used in any frequency band, so the circuit proposed in this paper has stronger scalability and wider applicability than other studies. Furthermore, the gain roll-off characteristics of transistors are mitigated by adding a negative feedback network. The PA proposed in this paper applies to the frequency of 1~3 GHz, and has a greater output capability over 40dBm in this band, with a harmonic rejection capability of −20 dBm or lower. The measured results show that the PA has perfect broadband performance and gain flatness while maintaining a high level and high efficiency of output. To sum up, this paper provides a new design solution for the realization of broadband PAs.

**Author Contributions:** Conceptualization, Z.L. and J.N.; methodology, Z.L.; software, Z.L.; validation, Z.L., J.N. and M.G.; formal analysis, Z.L.; investigation, Z.L.; resources, Z.L.; data curation, Z.L.; writing—original draft preparation, Z.L.; writing—review and editing, Z.L.; visualization, Z.L.; supervision, J.N. and Y.N.; project administration, M.G.; funding acquisition, J.N. All authors have read and agreed to the published version of the manuscript.

**Funding:** This research was funded by the National Natural Science Foundation of China（NO.61971210）

**Data Availability Statement:** The data that support the findings of this study are available from the corresponding author, [Zheng Li: lizheng20sui@163.com], upon reasonable request.

**Acknowledgments:** Thanks to Jingchang Nan for providing the experimental measurement environment.

**Conflicts of Interest:** The authors declare no conflict of interest.

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
