# Peer review of "A Broadband Power Amplifier Based on a Novel Filter Matching Network"

_electronics, doi:10.3390/electronics11223768_

Round 1
Reviewer 1 Report
Reviews
The topic “A broadband power amplifier based on a novel filter matching network” is presented in the paper using a new realization method of a broadband power amplifier based on a novel filter network. A novel matching network based on a band-pass filter has an excellent frequency selection function, which ensures an excellent matching characteristic in the aim band and generates strong harmonic suppression. But the following comments must be reported.
· Introduction section is too short, also, the citation styles are not the same in a whole document (not as per template). The third paragraph is not complete, and at the end of the introduction section, add the summary (organization) of your paper.
· Extensive grammar mistakes.
· Many abbreviations are not defined, such as RMID, HEMT, PAE, etc.
· In the figures, that are referring to someone, please cite in the caption of the figures, also citation should be in a proper sentence/paragraph.
· The parameters used in all the equations are not defined in a proper way, like Q, K, and D defining which parameters? Can you please explain?
· In equation 12, S21 is not defined.
· Differentiate the two letters, C, and c
· In line 105, justify this sentence “The simulation results of gain are shown in Figure 7.”
· Please use the curve markers, to differentiate the curves of the reflection coefficient in figures 7, 10, 13, and 14.
· Figure 16, shows the measured results of the power amplifier at different frequencies for AM-AM and AM-PM. What do you mean by the two terms AM-AM and AM-PM? Does anybody know about this two-term?
· Does “Harmonicre” exists?
· References are not in a proper format, also, ref. 1 and 2 are from the template.
Reviewer 2 Report
This paper describes the large bandwidth of the proposed PA based on filter matching network. This type of work using filter matching networks has been investigated for broadening the bandwidth of PA for a long time. To improve the quality of this work, the followings need to be clear.
1) Since the authors used GaN transistor for designing PA, they may have to clarify how to increase bandwidth of PA using GaN, along with presentation about which way is better fit for widening the GaN PA.
2) They also need to present how the proposed matching method is different from the previously published works in more detail.
3) In Table 2, used transistors (GaN?), supply voltage, type of class (class AB ? or others?), Bandwidth extension method and types of input signal (CW? Modulated signal?).
4) Many grammatical errors appear. So, they need to have a native English speaker proofread before submission.
5) Test for harmonic rejection should be presented in more detail with providing the used input signal type and a modulated input signal would be better for this test to decide the harmonic rejection or linearity.
6) Is the measured result for input matching provided?
Reviewer 3 Report
The paper addresses a single-ended power amplifier design. However, it does not provide significant new material. Beyond that, some sentences are very hard to follow due to grammatical errors. The comparison with previous works was not made correctly.
The sentence after Fig. 12 is not acceptable:
“Authors should discuss the results and how they can be interpreted from the perspective of previous studies and of the working hypotheses. The findings and their implications should be discussed in the broadest context possible. Future research directions may also be highlighted”
Specific comments:
What do the authors want to say with “… has excellent frequency select function, …” in the abstract?
The literature review in the introduction is not done correctly. There are much more relevant works that deal with broadband power amplifier design and with much better performance than the one obtained by the authors, such as:
N. Tuffy, L. Guan, A. Zhu and T. J. Brazil, "A Simplified Broadband Design Methodology for Linearized High-Efficiency Continuous Class-F Power Amplifiers," in IEEE Transactions on Microwave Theory and Techniques, vol. 60, no. 6, pp. 1952-1963, June 2012
Z. Zhang and Z. Cheng, "A Multi-Octave Power Amplifier Based on Mixed Continuous Modes," in IEEE Access, vol. 7, pp. 178201-178208, 2019
V. Carrubba et al., "On the Extension of the Continuous Class-F Mode Power Amplifier," in IEEE Transactions on Microwave Theory and Techniques, vol. 59, no. 5, pp. 1294-1303, May 2011
… and much more.
A HEMT, or any FET, does not have a physical RL, as is shown in Fig. 1.
What have authors done differently from the work of Dale E. Dawson in section II?
The authors did not define “r” in equation (5).
The authors should add a reference or explain why a Norton transformation would result in a constant load.
The sentence after Fig. 5 “the matching network can convert any impedance of any frequency to 50Ohm, and it can achieve the function only through third-order loops.” is very strong and erroneous. In high-power devices where C1 is very large and R1 is very small, and due to the large parasitic elements, the achievable bandwidth is limited.
The authors have used a feedback circuit to obtain a flat gain characteristic. However, there are works with the same device where the gain is flat without needing any feedback circuit. Moreover, if we look at Fig. 15, it is unclear if they used the feedback. I could not see the RCL circuit.
Then, they did not explain equation (12). In fact, it is very strange, since |S21| is changing in frequency, how could the authors have used it to calculate the resistance value that should not change in frequency?
The L and C values are not provided.
The authors did not explain how Fig.7 was obtained. Are these simulations for an unmatched device? If yes, the comparison does not make sense. The authors should compare the gain obtained considering the complete PA (with input and output matching work) with and without the feedback circuit.
The authors have just used one pair of input and output impedances to design the PA. The authors comment in the paragraph after table I that the optimum impedances of the device are changing in frequency, so they should select just one. This is not the correct procedure in a broadband design. First, the optimum intrinsic impedance may not be real (considering continuous classes of operation), and then, the impedance changes in frequency due to the Cds (and all the parasitics), which should be equivalent to the C1.
The conversion from lumped to distributed elements cannot be done in the entire frequency bandwidth. I doubt that the authors have only done a “slight tuned” in the final circuit.
Reviewer 4 Report
A broadband filter power amplifier is designed in this paper. There are the following suggestions for improvement.
(1) Proofread the full text by yourself, and delete repetitive and irrelevant contents, such as lines 154 to 165 on page 7.
(2) In Formula 12, | S21 | changes with frequency. How to take the value in design?
(3) Four S11 are written in Figure 10, two of which should be S21.
(4) In Figure 17, in order to prove the out of band suppression, the test results before 1G should also be displayed.
(5) The title of Figure 16 (b) should not be AM-AM.
(6) The innovation points of this paper should be further clarified.
Round 2
Reviewer 1 Report
Thanks for fixing the comments
Author Response
Thank you for your proposals. And you are our role model, and we will learn from you and continue to work hard.
Reviewer 2 Report
In the revised manuscript, something still needs to be improved as follows:
1) Still poor English writing since the revised sentences show some misused grammar and/or expressions.
2) The reviewer still does not understand why GaN device can provide wide bandwidth performance inherently after reading the added sentences they wrote.
3) For linearity test, they will have to use QAM signals to show out-of band rejection in the output power spectrum since they just now use a continuous waveform not a modulated waveform for the input signal. The proposed PA was designed for the wideband application so that they really need to test using the modulated input signal not the unmodulated continuous wave signal.
Reviewer 3 Report
Fig. 1 still has the same problem of including an RL resistor in the equivalent model, which does not make sense. RL is the impedance presented by the LC circuits and 50 ohm load and should not be presented in the equivalent circuit.
I still think the literature review in the introduction is not done properly. If the idea of the paper is to propose a design method for the output-matching network to improve bandwidth, why should the authors not compare with different techniques than the ones only related to filter synthetization?
Indeed, performance should not be the primary focus of a scientific manuscript. However, it indicates whether the developed theory is useful or not. Moreover, if the theory was already introduced in literature (even though for filter designing), it is only the application that is new, and, in that case, the performance is very relevant and should be compared with the state-of-the-art.
In Fig. 15, it seems that the authors have used a resistor in the output bias network. This creates a voltage drop and degrades the PA performance. Also, it is not clear how the input and output connectors are connected to the PCB.
I still believe that the negative feedback circuit is not needed. Although the |S21| of the unmatched device varies in frequency, the input matching work could be used to obtain a flat gain after designing the output-matching network to obtain the desired power and efficiency. Moreover, the explanation for using such negative feedback is unclear.
What was the input power used to obtain the measurements of Fig. 17b). The y-axis limits of this figure should be [-45 45].
Reviewer 4 Report
(1)It can be seen from Figure 5 that the proposed filter network can only deal with the conversion of real impedance to real impedance. How to deal with the complex impedance of the power amplifier?
(2)The title of Figure 10 is simulation results, but the simulation results and test results are written in the figure. Please clarify whether there are test results in it?
Round 3
Reviewer 3 Report
none